

# The fishing and natural mortality of large, piscivorous Bull Trout and Rainbow Trout in Kootenay Lake, British Columbia (2008–2013)

Joseph L. Thorley[1] and Greg F. Andrusak[2]

[1] Poisson Consulting, Nelson, British Columbia, Canada
[2] BC Fish and Wildlife Branch, Ministry of Forests, Lands and Natural Resource Operations, Victoria, British Columbia, Canada

## ABSTRACT

**Background**. Estimates of fishing and natural mortality are important for understanding, and ultimately managing, commercial and recreational fisheries. High reward tags with fixed station acoustic telemetry provides a promising approach to monitoring mortality rates in large lake recreational fisheries. Kootenay Lake is a large lake which supports an important recreational fishery for large Bull Trout and Rainbow Trout.

**Methods**. Between 2008 and 2013, 88 large ($\geq$500 mm) Bull Trout and 149 large ($\geq$500 mm) Rainbow Trout were marked with an acoustic transmitter and/or high reward ($100) anchor tags in Kootenay Lake. The subsequent detections and angler recaptures were analysed using a Bayesian individual state-space Cormack–Jolly–Seber (CJS) survival model with indicator variable selection.

**Results**. The final CJS survival model estimated that the annual interval probability of being recaptured by an angler was 0.17 (95% CRI [0.11–0.23]) for Bull Trout and 0.14 (95% CRI [0.09–0.19]) for Rainbow Trout. The annual interval survival probability for Bull Trout was estimated to have declined from 0.91 (95% CRI [0.76–0.97]) in 2009 to just 0.46 (95% CRI [0.24–0.76]) in 2013. Rainbow Trout survival was most strongly affected by spawning. The annual interval survival probability was 0.77 (95% CRI [0.68–0.85]) for a non-spawning Rainbow Trout compared to 0.41 (95% CRI [0.30–0.53]) for a spawner. The probability of spawning increased with the fork length for both species and decreased over the course of the study for Rainbow Trout.

**Discussion**. Fishing mortality was relatively low and constant while natural mortality was relatively high and variable. The results indicate that angler effort is not the primary driver of short-term population fluctuations in the Rainbow Trout abundance. Variation in the probability of Rainbow Trout spawning suggests that the spring escapement at the outflow of Trout Lake may be a less reliable index of abundance than previously assumed. Multi-species stock assessment models need to account for the fact that large Bull Trout are more abundant than large Rainbow Trout in Kootenay Lake.

Corresponding author
Joseph L. Thorley,
joe@poissonconsulting.ca

## INTRODUCTION

Estimates of fishing ($F$) and natural mortality ($M$) are important for understanding, and ultimately managing, commercial and recreational fisheries (*Hilborn, 1992*; *Walters & Martell, 2004*). However separating the contributions of $F$ and $M$ to the total mortality ($Z$) of a fish population is challenging (*Quinn & Deriso, 1999*). In the case of large lake recreational fisheries, a promising approach (*Bacheler et al., 2009*) combines a high reward tagging program to provide information on $F$ (*Pollock et al., 2001*) with an acoustic telemetry study to provide information on $M$ (*Pollock, Jiang & Hightower, 2004*).

Kootenay Lake is a large (388 km$^2$) enhanced waterbody in the upper Columbia River drainage that supports a 200,000 rod hour per year recreational fishery (*Andrusak & Andrusak, 2012*) primarily targetting large ($\geq$500 mm) Bull Trout (*Salvelinus confluentus*) and Rainbow Trout (*Onchorhynchus mykiss*). The Bull Trout are an adfluvial population that spawn in the many tributaries of Kootenay Lake (*Andrusak & Andrusak, 2014*). The large Rainbow Trout, which are often referred to as Gerrard Rainbow Trout (*Hartman & Galbraith, 1970*), represent a genetically distinct (*Keeley, Parkinson & Taylor, 2007*) piscivorous ecotype (*Keeley, Parkinson & Taylor, 2005*). Both the Bull Trout and Rainbow Trout can reach trophy size ($\geq$9 kg) in Kootenay Lake.

Kootenay Lake has experienced several anthropogenic shifts in its nutrient loading over the past six decades (*Northcote, 1973*). In the 1950s and 1960s it was eutrophic due to nutrient pollution. Then, in the 1970s, pollution reduction and the retention of nutrients in the recently constructed upstream Duncan and Libby reservoirs caused a long-term decline in lake productivity (*Daley et al., 1980*). As a result Kokanee (*Oncorhynchus nerka*), which provide the primary food source for the piscivorous trout (*Andrusak & Parkinson, 1984*), also declined. In response, the North Arm of Kootenay Lake began to be fertilized in 1992 (*Schindler et al., 2014*). However, between 1997 and 2000, the fertilizer inputs were temporarily experimentally reduced before returning to pre-1997 levels in 2001. Most recently, in 2004, the fertilization program was expanded to also include the South Arm (*Schindler et al., 2014*).

Since the late 1980s, the large piscivorous Rainbow Trout in Kootenay Lake have exhibited population cycles of approximately seven to eight years in duration (*Kurota et al., 2016*). Possible explanations for the short-term cycles include angler effort or Kokanee abundance (*Kurota et al., 2016*). If angling effort is the primary driver then $F$ would be expected to be varying through time. Conversely if Kokanee abundance or another biological factor is responsible then $M$ should be variable. In contrast, relatively little is known about the status of the Bull Trout population; although preliminary analysis of redd counts suggest a decline in Bull Trout spawner abundance from 2009 to 2013 (*Andrusak & Andrusak, 2014*).

Here we document a combined tag-telemetry study to estimate the natural and fishing mortality of the large ($\geq$500 mm) Bull Trout and Rainbow Trout in Kootenay Lake. A secondary objective was to test whether $F$ and/or $M$ were changing through time. The study commenced in 2008 with a one year pilot program that was followed by three more years (2009–2011) of high reward tagging and acoustic telemetry and then a final two years (2012–2013) of high reward tagging only. The data were analysed using a Bayesian

individual state-space formulation (*Royle, 2008*; *Kéry & Schaub, 2011*) of the Cormack–Jolly–Seber (CJS) (*Cormack, 1964*; *Jolly, 1965*; *Seber, 1965*) survival model to take into account acoustic detection probabilities, angler recaptures, spawning state and growth. Model selection was achieved using indicator variable selection (*Hooten & Hobbs, 2015*). In 2011 a creel survey was conducted on Kootenay Lake (*Andrusak & Andrusak, 2012*). The recapture rates from the current study are combined with the effort and catch estimates from the creel survey to estimate the catchability and density for both populations.

## METHODS

### Study area

The main body of Kootenay Lake (116.905°E 49.635°S) is a large (388 km$^2$), long (109 km), narrow (4 km), deep (mean depth c. 100 m) volume of water in the upper Columbia River drainage of southeastern British Columbia (Fig. 1). The Duncan-Lardeau River system feeds the North Arm while the Kootenay River feeds the South Arm (Fig. 1). The outlet of the main lake forms the upper end of the shallow West Arm which includes a series of narrow riverine sections. Lake levels and temperatures are influenced by the upstream Duncan and Libby Dams and by the downstream Corra Linn Dam (*Hamblin & McAdam, 2003*). For more information on Kootenay Lake see *Daley et al. (1980)*, *Schindler et al. (2014)* and *Kurota et al. (2016)*.

### Acoustic receivers

The study was made possible by the pre-existence of an acoustic receiver array. The array, which was supplemented so that it included a total of 25 VR2(W) 81 kHz Vemco® acoustic receivers in Kootenay Lake (Fig. 1), was originally designed to track juvenile White Sturgeon (*Acipenser transmontanus*) (*Neufeld & Rust, 2009*). The array has also been used to track fluvial Bull Trout (*Paragamian & Walters, 2011*) and juvenile Burbot (*Lota lota*) (*Stephenson et al., 2013*). An additional three VR2(W) 81 kHz Vemco® acoustic receivers were deployed each year in the Lardeau River at the outflow of Trout Lake (Fig. 1) in April and May to detect spawning Rainbow Trout (*Irvine, 1978*; *Irvine, Baxter & Thorley, 2013*).

The lake receivers were held in position by an anchor and a high pressure trawl float about 30 m below the surface. The receivers themselves were orientated in a downward direction at a depth of about 45 m. For more information on lake receiver deployment see *Neufeld & Rust (2009)*. At the outflow of Trout Lake the three receivers were mounted to bankside submerged structures and orientated across the channel.

### Acoustic transmitters

Between 2008 and 2011, Bull Trout and Rainbow Trout with a fork length ≥ 500 mm were tagged with a V13-1L or V13-1LP 81 kHz Vemco® acoustic transmitter. The V13-1L transmitters had a diameter of 13 mm, length of 36 mm and mass of 11 g while the V13P-1L transmitters had a diameter of 13 mm, length of 45 mm and mass of 12 g. In the pilot year (2008) the V13-1L transmitters had a nominal pulse frequency of 60 s (30–90 s) which resulted in a tag life of 455 days. In all subsquent years the nominal pulse rate was 120 s

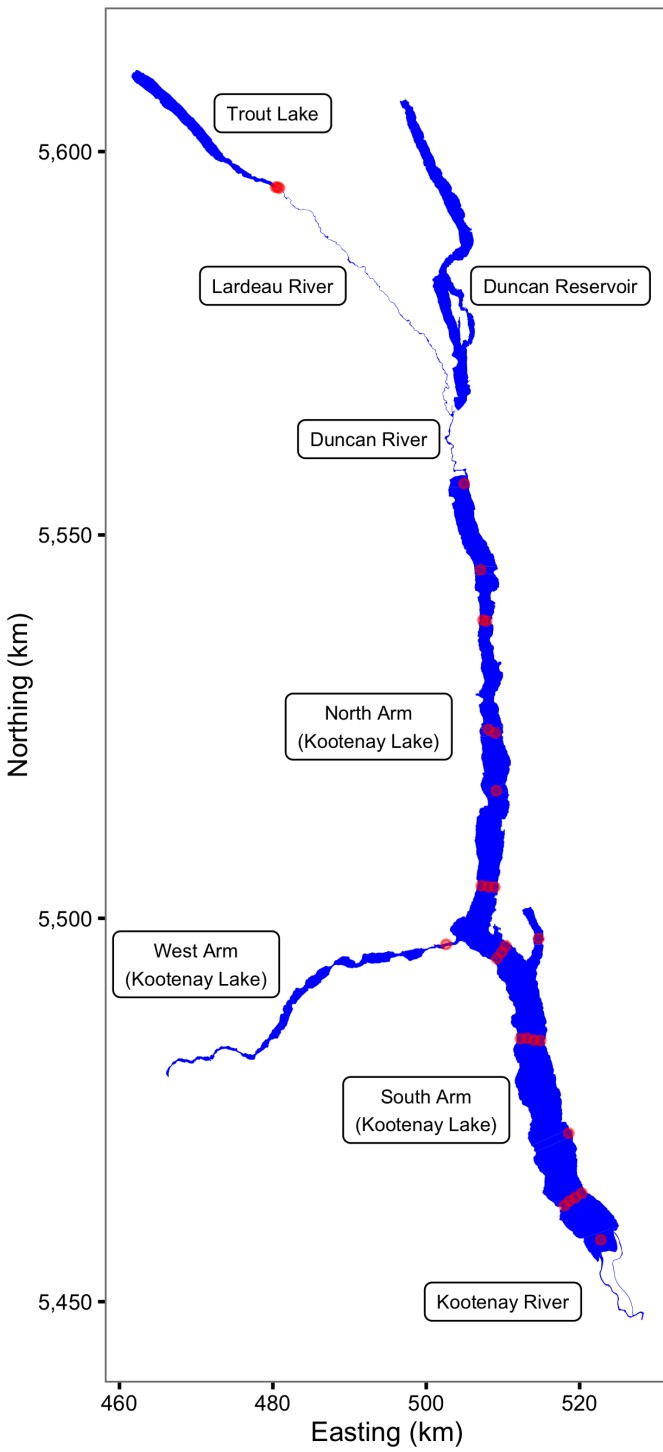

**Figure 1** **The study area including Kootenay Lake.** The locations of the acoustic receivers are indicated by red circles. Spatial information licensed under the Open Government License of British Columbia.

(60–180 s) which resulted in a tag life of 1,239 days for the V13-1L transmitters and 830 days for the V13P-1L transmitters. Recovered transmitters with a remaining tag life of 730 days or more were redeployed in new fish. Approximately six percent of the acoustically tagged fish were tracked using a recovered transmitter.

## Reward tags

All fish were tagged with an orange $100 reward anchor tag (*Pollock et al., 2001*). Following the pilot year, all but one Bull Trout and two Rainbow Trout were also tagged with a green $10 reward anchor tag, a blue standard anchor tag or another $100 reward anchor tag to provide information on tag loss (*Fabrizio et al., 1999*). The anchor tags were Floy® FD-68B T-bar anchor tags with 16 mm monofilament and 44 mm tubing. The anchor tags were attached below the dorsal fin on opposite sides using a Floy® Mark III regular pistol grip tagging gun.

To maximize anchor tag reporting (*Pollock et al., 2001*) the anchor tags displayed a government telephone number. The orange and green reward tags also displayed the text "REWARD $100" or "REWARD $10," respectively. The reward tagging program was advertised on posters and in the local media (newspapers, radio and television). Rewards were only paid for returned anchor tags. Anglers were interviewed by phone to determine the date and location of recapture.

## Fish capture and handling

Fish were caught by trolling lures with barbless hooks behind a boat. Angled fish were quickly played to the side of the boat, netted, unhooked and placed into a 150 L cooler of aerated fresh water where they were allowed to recover for approximately 20 min. Only individuals with a fork length ≥500 mm that had not suffered hooking damage, were not heavily bleeding and were able to maintain equilibrium were acoustically and/or reward tagged.

Once recovered, angled fish were transferred to a second 150 L cooler of aerated fresh lake water and clove oil (emulsified in ethanol) at a concentration of 50 mg/L (*Anderson, McKinley & Colavecchia, 1997*; *Prince & Powell, 2000*). Fish were then held in the second cooler for approximately 150 s until they exhibited total loss of equilibrium and no longer responded to squeezing the caudal peduncle (*Gutowsky et al., 2013*; *Gutowsky et al., 2016*).

The anaesthetised fish was then immediately placed upside down on a V-board where its gills were gently irrigated using a wash bottle filled with water from the second cooler. The acoustic tag and surgical equipment had been sterilised previously by soaking for approximately 20 min in 70% ethanol while the fish was recovering from capture. The acoustic tag and surgical equipment were rinsed with sterile saline solution prior to surgery to remove the ethanol. A scapel blade (number 11) was used to make an incision of approximately 50 mm in length midway between the pelvic and pectoral fish, close to and parallel to the mid-ventral line. After insertion of the acoustic tag, an Ethicon® suture with 700 mm of 1.5 metric polydioxanone monofilament and a 19 mm 3/8 circle reverse cutting needle was used to close the incision with two or three interrupted stitches. Midway through stitching the irrigation wash bottle was replaced with one filled with fresh lake water.

Following surgical implantation of an acoustic transmitter, each fish was measured (± 1 mm) and returned to the recovery cooler. While the fish was recovering it was anchor

tagged and adipose clipped. All tagged fish were allowed to recover from surgery for approximately 20 min before being released within 500 m of the point of capture. In 2012 and 2013, fish were anchor tagged only and so were not anesthetised or adipose clipped. Fish were obtained under scientific collection permits (CB08-43988, CB09-53420, CB10-61021, CB11-69505, CB12-76723) issued by the British Columbia Ministry of Forests Lands and Natural Resource Operations (MFLNRO).

## Data manipulation
### Raw data
The acoustic receiver VRL download files were processed using Vemco® VUE software to calculate the number of hourly detections of each transmitter by each acoustic receiver. The reported recaptures were recorded by MFLNRO staff and the rewards administered by the Freshwater Fish Society of British Columbia (FFSBC). Based on lake geomorphology and the locations of the receivers (Fig. 1) the study area was divided into 30 sections (Fig. 2).

### Hourly receiver data
The hourly detection, receiver deployment, fish capture and recapture data sets and sectional shapefiles were manipulated using R version 3.3.2 (*R Core Team, 2015*). During data manipulation, two or less detections of a fish at a receiver in an hour or any detections after the expected tag life were discarded. The resultant clean and tidy (*Wickham, 2014*) data sets were bundled together in an R data package called klexdatr (Article S1).

### Daily sectional data
The hourly receiver detection data were then aggregated into daily sectional detection data using the lexr R package (Article S1). During the aggregation process, acoustically tagged individuals that 30 days after release were no longer detected or were only detected at a single section were classified as post-release mortalities and were excluded (*Hightower, Jackson & Pollock, 2001*). Detections of the same fish in more than one section in the same day (7% of the total) were resolved in favour of the section with the most detections. Ties were broken by choosing the section with the least receivers and if still tied the smaller section (sectional areas were unique). The total percent receiver coverage of each section in each interval was calculated (Fig. 3) assuming a detectional radius of 500 m (*Gutowsky et al., 2013*; *Gutowsky et al., 2016*; *Huveneers et al., 2015*).

### Seasonal sectional data
The lexr package was then used to group the daily data into seasonal periods for analysis. Following *Gutowsky et al. (2013)*, the seasons were winter (January–March), spring (April–June), summer (July–September) and autumn (October–December). The final data included logical matrices indicating for each fish in each seasonal period whether it was monitored (active acoustic transmitter) and/or reported (recaptured) and whether it moved (detected in two or more sections) and/or spawned.

Spawning in Bull Trout was identified by a hiatus in detections for at least four weeks during August and September when the fish was deemed to have moved out of the main lake (*O'Brien, 2001*; *Andrusak & Andrusak, 2014*). For Rainbow Trout spawning was identified

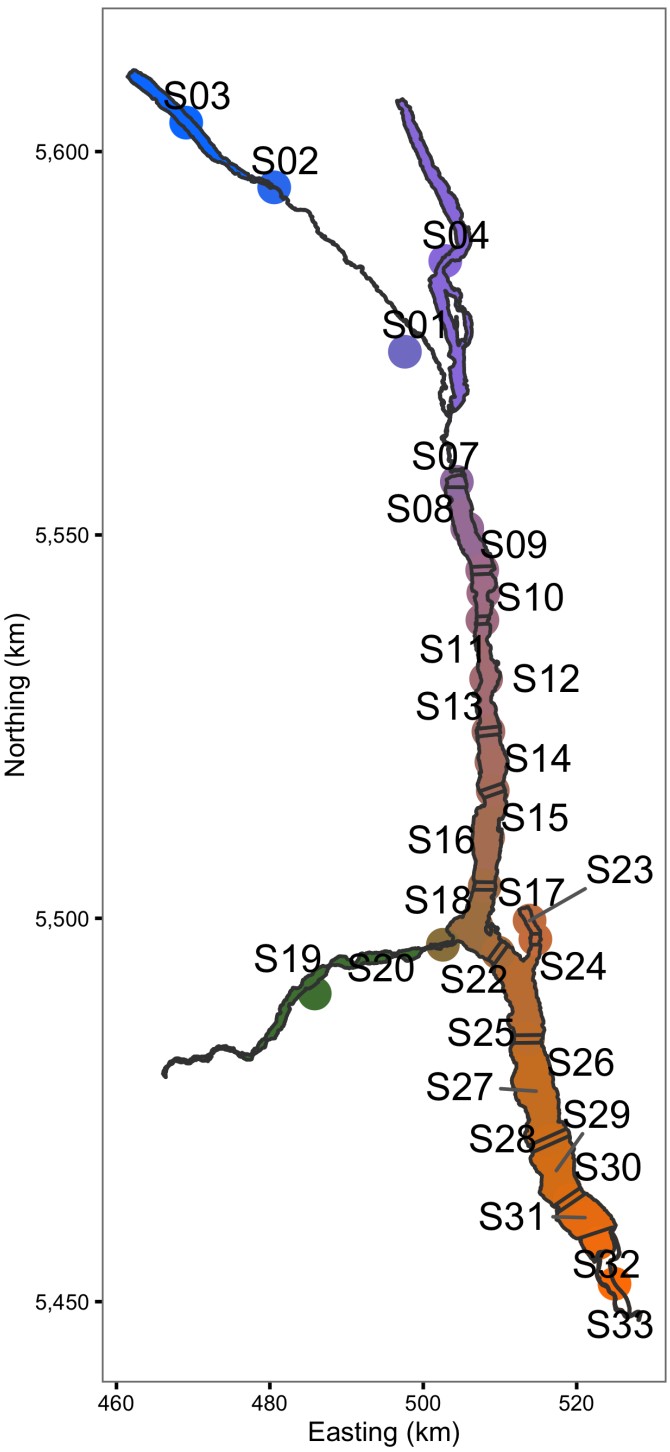

**Figure 2 The study area including Kootenay Lake by color-coded section.** The same color-coding scheme is used in the following two figures. Spatial information licensed under the Open Government License of British Columbia.

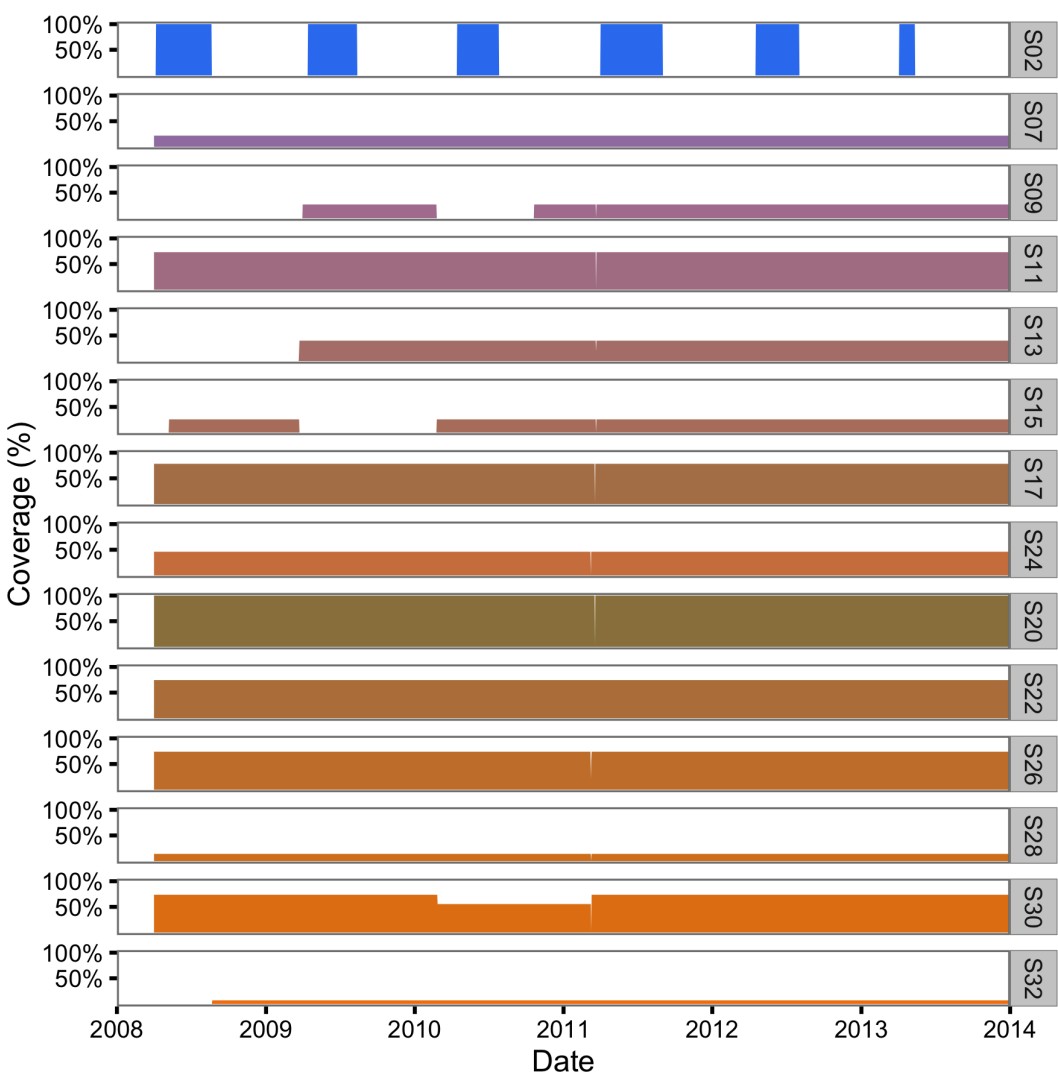

**Figure 3  Daily acoustic receiver coverage in the study area by color-coded section.**

by either a detection in April or May at the outflow of Trout Lake (section S02) or a hiatus in detections for three or more weeks during April and May (*Irvine, 1978*; *Irvine, Baxter & Thorley, 2013*) with the bracketing detections occuring in the top 9% of Kootenay Lake (sections S07–S09).

## Statistical analysis
### CJS model

The probabilities of being recaught (and reported) by an angler versus dying of other causes were estimated from the seasonal data using an individual state-space (*Royle, 2008*; *Kéry & Schaub, 2011*) Cormack–Jolly–Seber (CJS) (*Cormack, 1964*; *Jolly, 1965*; *Seber, 1965*) survival model. In the individual state-space formulation of the CJS model, the *i*th individual is alive when it is initially tagged at time period $f_i$, e.g.,

$$z_{i,f_i} = 1. \tag{1}$$

Its latent state (alive versus dead) at subsequent periods is modelled as the outcome of a series of Bernoulli trials

$$z_{i,t+1} \sim Bernoulli(z_{i,t} \cdot \phi_{i,t}) \tag{2}$$

where $\phi_{i,t}$ is the predicted survival for the $i$th fish in the $t$th time period. An individual that is alive at period $t$ also has a probability $\rho_{i,t}$ of being recaptured, i.e.,

$$y_{i,t} \sim Bernoulli(z_{i,t} \cdot \rho_{i,t}). \tag{3}$$

### Base model

In the base model of the current study, $\rho_{i,t}$ is the probability of being recaptured (and reported) by an angler. To reduce the number of necessary assumptions, reported recaptures are excluded from the analysis for all subsequent time periods, i.e., subsequent detections or recaptures of any released individuals were ignored. In addition, a living individual with an active transmitter ($T_{i,t} = 1$) also has a probability ($\delta_{i,t}$) of being detected moving ($m_{i,t}$) between sections by the receiver array

$$m_{i,t} \sim Bernoulli(z_{i,t} \cdot T_{i,t} \cdot \delta_{i,t}). \tag{4}$$

Finally, in the spawning season ($S_{i,t} = 1$) a fish has a probability $\kappa_{i,t}$ of entering the state of spawning ($x_{i,t} = 1$) for the period

$$x_{i,t} \sim Bernoulli(z_{i,t} \cdot S_{i,t} \cdot \kappa_{i,t}). \tag{5}$$

In the base model, the terms $\phi_{i,t}$, $\rho_{i,t}$, $\delta_{i,t}$ and $\kappa_{i,t}$, which represent probabilities, are specified by four parameters, i.e.,

$$logit(\phi_{i,t}) = \beta_{\phi 0}, \tag{6}$$
$$logit(\rho_{i,t}) = \beta_{\rho 0}, \tag{7}$$
$$logit(\delta_{i,t}) = \beta_{\delta 0} \tag{8}$$

and

$$logit(\kappa_{i,t}) = \beta_{\kappa 0} \tag{9}$$

where

$$logit(p) = \log\left(\frac{p}{1-p}\right). \tag{10}$$

### Full model

The full model extends the base model through the inclusion of seven additional parameters ($\beta_{\kappa L}$, $\beta_{\phi x}$, $\beta_{\delta S}$, $\beta_{\kappa Y}$, $\beta_{\rho Y}$, $\beta_{\phi Y}$ and $\beta_{\delta Y}$). The first parameter ($\beta_{\kappa L}$) allows the log odds probability of spawning to vary with the calculated fork length ($L_{i,t}$)

$$logit(\kappa_{i,t}) = \beta_{\kappa 0} + \beta_{\kappa L} \cdot L_{i,t} \tag{11}$$

while the second ($\beta_{\phi x}$) allows the log odds survival to vary with spawning

$$logit(\phi_{i,t}) = \beta_{\phi 0} + \beta_{\phi x} \cdot x_{i,t}. \tag{12}$$

The fork lengths were calculated based on the measured length at capture $L_{i,f_i}$ plus the length increment expected under a Von Bertalanffy Growth Curve (*Walters & Martell, 2004*)

$$L_{i,t} = L_{i,f_i} + (L_\infty - L_{i,f_i})(1 - e^{-0.25 \cdot k(t-f_i)}) \tag{13}$$

where the 0.25 in the exponent adjusts for the fact that there are four time periods (seasons) per year. The calculation assumes a $L_\infty$ of 1,000 mm with a $k$ of 0.13 for Bull Trout and a $k$ of 0.19 (*Andrusak & Thorley, 2014*) for Rainbow Trout.

The third additional parameter ($\beta_{\delta S}$) allows the log odds probability of being detected moving to vary with the spawning season

$$logit(\delta_{i,t}) = \beta_{\delta 0} + \beta_{\delta S} \cdot S_{i,t} \tag{14}$$

while the last four parameters ($\beta_{\kappa Y}$, $\beta_{\rho Y}$, $\beta_{\phi Y}$ and $\beta_{\delta Y}$) allow the probability of spawning, recapture, survival, and detection moving between sections to vary with the standardised year ($Y_{i,t}$), i.e,

$$logit(\kappa_{i,t}) = \beta_{\kappa 0} + \beta_{\kappa Y} \cdot Y_{i,t}. \tag{15}$$

The parameters in the full model are defined in Table 1.

*Parameter estimation.* The parameters were estimated using Bayesian methods (*Ntzoufras, 2009*; *Kéry, 2010*; *Kéry & Schaub, 2011*). The prior distribution for each parameter was a normal distribution with a mean of 0 and a standard deviation of 3, i.e.,

$$\beta_{\kappa 0} \sim Normal(0,3). \tag{16}$$

The posterior distributions of the parameters were estimated using a Monte Carlo Markov Chain (MCMC) algorithm. To avoid non-convergence of the MCMC process, five chains were run, starting at randomly selected initial values. Each chain was run for at least $10^5$ iterations with the first half of the chain discarded for burn-in followed by further thinning to leave a grand total of 10,000 samples. Convergence was confirmed by ensuring that $\hat{R} \leq$ 1.05 for each of the parameters in the model (*Brooks & Gelman, 1998*; *Kéry & Schaub, 2011*).

The vagueness of the priors was assessed by a sensitivity analysis (*Kéry & Schaub, 2011*). More specifically, the model was refitted with the prior distribution for each parameter a normal distribution with a mean of 0 and a standard deviation of 6. The sensitivity of the posteriors to the change in priors was assessed using $\hat{R}$. Although developed to quantify the convergence between chains, $\hat{R}$ can also be used as a metric of the convergence between models by combining all the samples for each model. The vagueness of the priors was confirmed by ensuring that the between model $\hat{R}$ values were all $\leq$1.2. A higher minimum $\hat{R}$ value was accepted for the between model (than the within model convergence) because

**Table 1   Descriptions for each of the parameters in the full CJS model.**

| Parameter | Description |
|---|---|
| Y | The standardised year. |
| FL | The calculated fork length. |
| L | (FL − 600) / 100 |
| $\beta_{\delta 0}$ | The log odds seasonal probability of being detected moving among sections. |
| $\beta_{\delta S}$ | The effect of spawning season on $\beta_{\delta 0}$. |
| $\beta_{\delta Y}$ | The effect of Y on $\beta_{\delta 0}$. |
| $\beta_{\kappa 0}$ | The log odds probability of spawning. |
| $\beta_{\kappa L}$ | The effect of L on $\beta_{\kappa 0}$. |
| $\beta_{\kappa Y}$ | The effect of Y on $\beta_{\kappa 0}$. |
| $\beta_{\phi 0}$ | The log odds seasonal probability of surviving. |
| $\beta_{\phi \kappa}$ | The effect of spawning on $\beta_{\phi 0}$. |
| $\beta_{\phi Y}$ | The effect of Y on $\beta_{\phi 0}$. |
| $\beta_{\rho 0}$ | The log odds seasonal probability of being recaptured. |
| $\beta_{\rho Y}$ | The effect of Y on $\beta_{\rho 0}$. |
| $\gamma_{\delta S}$ | The selection probability for $\beta_{\delta S}$. |
| $\gamma_{\delta Y}$ | The selection probability for $\beta_{\delta Y}$. |
| $\gamma_{\kappa L}$ | The selection probability for $\beta_{\kappa L}$. |
| $\gamma_{\kappa Y}$ | The selection probability for $\beta_{\kappa Y}$. |
| $\gamma_{\phi \kappa}$ | The selection probability for $\beta_{\phi \kappa}$. |
| $\gamma_{\phi Y}$ | The selection probability for $\beta_{\phi Y}$. |
| $\gamma_{\rho Y}$ | The selection probability for $\beta_{\rho Y}$. |

the purpose of the test was to confirm that the posteriors were not unduly altered by the change in the priors, i.e., except in the case of simple models with lots of data it is not realistic to expect that a substantial change in the priors will have no effect at all on any of the posteriors. The chosen $\hat{R}$ value of 1.2 is within the accepted range of values used to confirm the convergence of chains within a single model (*Kéry & Schaub, 2011*).

The reported point estimates are the mean and the 95% credible intervals (CRIs) are the 2.5 and 97.5% quantiles (*Gelman, 2014*). Unless stated otherwise all presented and plotted estimates are for a representative individual which in the current study was judged to be a 650 mm non-spawner in the autumn of 2011.

### Model selection

Model selection was achieved by using indicator variable selection (*Hooten & Hobbs, 2015*) to estimate probabilities for the inclusion of the seven additional parameters from the full model in a final model. More specifically, "stochastic search variable selection" (*George & McCulloch, 1993*; *Hooten & Hobbs, 2015*) was used to assign an indicator variable to each parameter such that

$$\gamma_{\kappa L} \sim Bernoulli(1/2) \tag{17}$$

$$\beta_{\kappa L} \sim \gamma_{\kappa L} \cdot Normal(0,3) + (1 - \gamma_{\kappa L}) Normal(0,0.03). \tag{18}$$

In other words, if the indicator is 1 then the parameter is drawn from the standard vague prior, otherwise the parameter is drawn from a prior that is so constrained that its value is effectively zero. The values of $\gamma_{\rho Y}$ and $\gamma_{\phi Y}$ indicate the support for a change in $F$ and $M$, respectively, over the course of the study.

*Software.* The analyses were performed using R version 3.3.2 (*R Core Team, 2015*), JAGS 4.2.0 (*Plummer, 2003*) and the klexr R package (*Article S1*), which was developed specifically for this paper.

### Model assumptions

The validity of a statistical model's estimates depends on the extent to which its assumptions are violated (*Kéry & Schaub, 2011*). Like the standard CJS model, the modified form used in the current analysis assumes that

1. The tagged individuals are a random sample from the population.
2. There are no effects of tagging.
3. There is no loss of anchor or acoustic tags.
4. All individuals are correctly identified.
5. There is no emigration out of the main lake.
6. There are no unmodelled individual differences in the probability of recapture in each time interval.
7. There are no unmodelled individual differences in the probability of survival in each time interval.
8. The fate of each individual is independent of the fate of any other individual.
9. The sampling periods are instantaneous and all individuals are immediately released.
   In addition, the modified form used in the current study also assumes that

10. There are no unmodelled individual differences in the probability of a living individual with an active acoustic transmitter being detected moving among sections in each time period.
11. Anglers report all anchor tags.
12. Spawning events are correctly identified.
13. Growth increments are correctly calculated.

The extent to which each assumption may have been violated was assessed based on the literature and, when available, auxilary data. In particular, the random nature of the population sample was assessed by plotting the spatial distribution of captures, daily detections and recaptures. The extent of any outmigration was evaluated from the proportion of acoustically tagged individuals last detected at the top of the North Arm (S07), bottom of the South Arm (S32) and at the upper end of the West Arm (S20). Only individuals that were last detected at least 120 days before the end of their transmitter life and were not subsequently recaptured were considered to be possible outmigrants. Finally, anchor tag loss was estimated from the proportion of double-tagged individuals that had lost an anchor tag at recapture using a simple Bayesian zero-truncated binomial tag-loss model (*Fabrizio et al., 1999*).

Table 2  **The number of captured Bull Trout by year and the value of the second anchor tag.** All in cases the first anchor tag was a $100 reward tag. With the exception of 2012 and 2013, the fish were also acoustically tagged.

| Year | $100 | $10 | $0 | No |
|------|------|-----|-----|-----|
| 2008 | 0 | 0 | 0 | 0 |
| 2009 | 0 | 21 | 1 | 0 |
| 2010 | 0 | 26 | 0 | 0 |
| 2011 | 0 | 21 | 0 | 0 |
| 2012 | 0 | 12 | 0 | 1 |
| 2013 | 0 | 6 | 0 | 0 |

Table 3  **The number of captured Rainbow Trout by year and the value of the second anchor tag.** All in cases the first anchor tag was a $100 reward tag. With the exception of 2012 and 2013, the fish were also acoustically tagged.

| Year | $100 | $10 | $0 | No |
|------|------|-----|-----|-----|
| 2008 | 1 | 0 | 0 | 17 |
| 2009 | 0 | 30 | 1 | 0 |
| 2010 | 0 | 29 | 0 | 0 |
| 2011 | 1 | 34 | 0 | 1 |
| 2012 | 0 | 26 | 0 | 0 |
| 2013 | 0 | 8 | 0 | 1 |

# RESULTS

## Fish captures and recaptures
### Captures
Between 2008 and 2013, 88 large ($\geq$500 mm) Bull Trout and 149 large ($\geq$500 mm) Rainbow Trout were marked with at least one $100 reward anchor tag (Tables 2 and 3). A total of 69 of the Bull Trout and 114 of the Rainbow Trout were also acoustically tagged. Most Bull Trout were caught in the central portion of the main lake or the top of the North Arm (Fig. S1) while most Rainbow Trout were caught in the central portion of the main lake (Fig. S2).

### Post-release mortalities
Of the acoustically tagged individuals five (7%) of the Bull Trout and 28 (25%) of the Rainbow Trout were no longer detected moving between sections 30 days after release. They were deemed to be post-release mortalities and were excluded from the data. The daily detections are plotted by section and date in Fig. 4 for the remaining 83 Bull Trout and 121 Rainbow Trout.

### Detections
The spatial distribution of the daily detections indicates that fish use of the main lake was relatively evenly distributed for both Bull Trout (Fig. S3) and Rainbow Trout (Fig. S4). A grand total of 27 acoustically tagged Bull Trout, which were not subsequently recaught, were last detected moving in the main lake 120 days before the end of their transmitter life

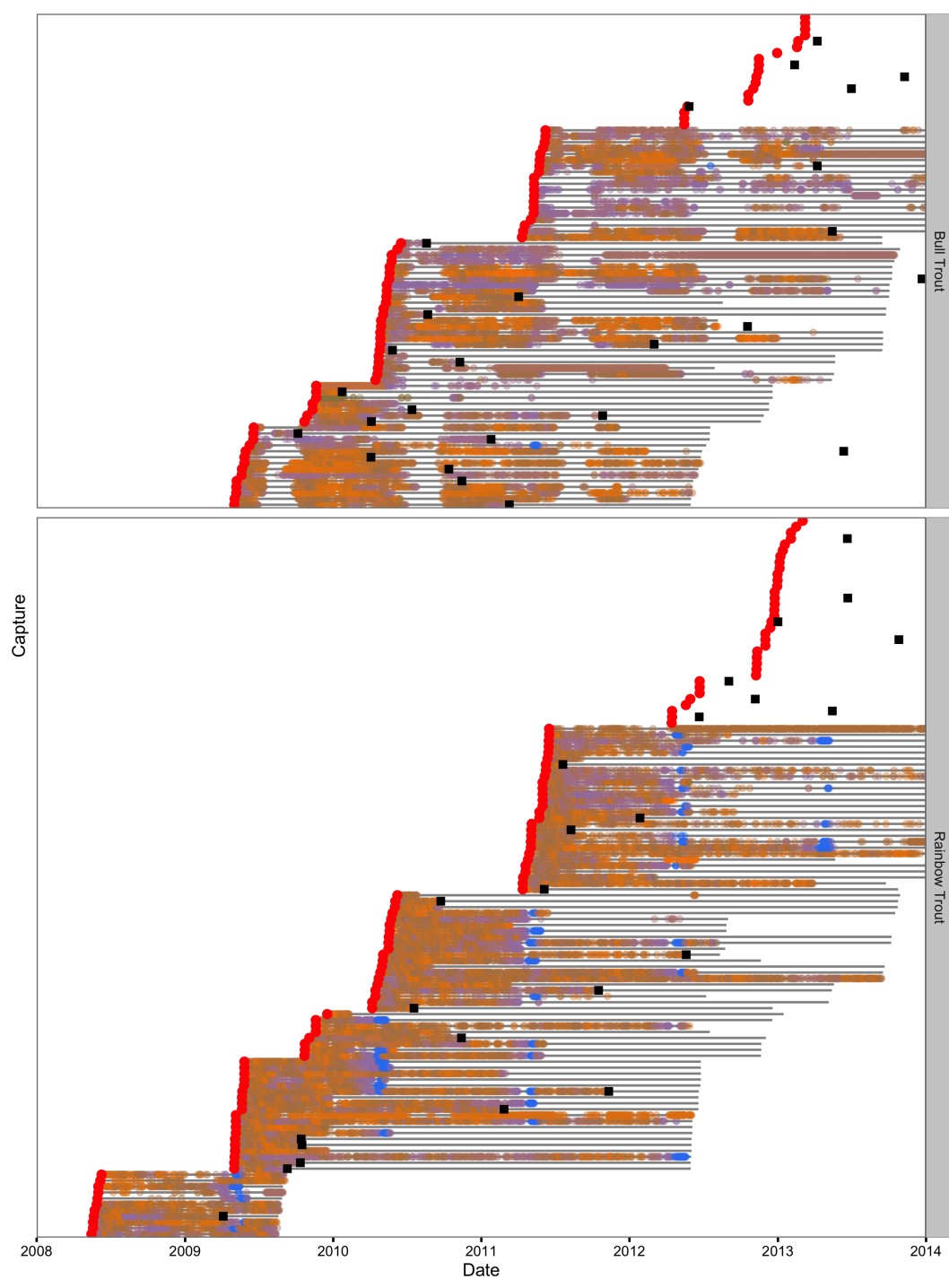

**Figure 4 Daily acoustic detections in the study area by fish, species and color-coded section.** Captures are indicated by the red circles, recaptures by the black squares and transmitter life by the horizontal grey lines.

**Table 4  The number of Bull Trout last detected at each section by season.** Only individuals that were last detected at least 120 days before the end of their transmitter life and were not subsequently recaptured are included.

| Section | Winter | Spring | Summer | Autumn |
| --- | --- | --- | --- | --- |
| S07 | 0 | 6 | 1 | 0 |
| S09 | 0 | 1 | 0 | 2 |
| S11 | 0 | 1 | 1 | 1 |
| S13 | 0 | 2 | 1 | 0 |
| S15 | 1 | 0 | 0 | 0 |
| S17 | 0 | 2 | 0 | 1 |
| S24 | 1 | 0 | 0 | 1 |
| S20 | 0 | 0 | 0 | 1 |
| S22 | 1 | 0 | 0 | 0 |
| S26 | 0 | 1 | 0 | 0 |
| S30 | 0 | 1 | 0 | 1 |

**Table 5  The number of Rainbow Trout last detected at each section by season.** Only individuals that were last detected at least 120 days before the end of their tranmitter life and were not subsequently recaptured are included.

| Section | Winter | Spring | Summer | Autumn |
| --- | --- | --- | --- | --- |
| S02 | 0 | 11 | 0 | 0 |
| S07 | 0 | 7 | 0 | 0 |
| S09 | 1 | 0 | 0 | 0 |
| S11 | 0 | 3 | 1 | 1 |
| S17 | 0 | 0 | 1 | 1 |
| S24 | 0 | 0 | 1 | 1 |
| S20 | 1 | 2 | 1 | 0 |
| S22 | 1 | 2 | 0 | 1 |
| S26 | 2 | 2 | 0 | 0 |
| S28 | 0 | 0 | 1 | 0 |
| S30 | 2 | 0 | 1 | 1 |
| S32 | 0 | 3 | 0 | 0 |

(Table 4). Of the 27 Bull Trout, seven (26%) were last detected moving at section S07, one (4%) at S20 and zero (0%) at S32. For Rainbow Trout, the grand total was 48 individuals (Table 5), of which seven (15%) were last detected at section S07, four (8%) at S20 and three (6%) at S32.

### Recaptures

A total of 26 of the 83 Bull Trout and 24 of the 121 Rainbow Trout were reported to have been recaught by an angler (Fig. 4). In the main lake, most of the Bull Trout were recaught in the central portion or the top of the North Arm (Fig. S5). In contrast the Rainbow Trout recaptures were relatively evenly distributed throughout the main lake with the exception of the most northerly sections of the North Arm and the most southerly sections of the South Arm (Fig. S6).

Of the 25 Bull Trout recaptures that had been initially tagged with a $100 and a $10 tag, 18 had both anchor tags, three had just the $100 tag and four had just the $10 tag. For the 22 equivalent Rainbow Trout recaptures, 20 had both anchor tags, zero had just the $100 tag and two had just the $10 tag. Analysis of the number of individuals with one versus two anchor tags using the simple Bayesian tag-loss model estimates the probability of a single-tagged individual losing its anchor tag between release and recapture to be 0.18 (95% CRI [0.08–0.32]) for Bull Trout and 0.07 (95% CRI [0.01–0.17]) for Rainbow Trout. The corresponding probability of a double-tagged individual losing both its anchor tags (assuming the fates of both tags are independent) is 0.04 (95% CRI [0.01–0.10]) and 0.006 (95% CRI [0.000–0.027]), for Bull Trout and Rainbow Trout, respectively.

## Parameter estimates and model selection
### Parameter estimation
Model convergence was confirmed by an $\hat{R} \leq 1.02$ for Bull Trout and $\leq 1.02$ for Rainbow Trout. The vagueness of the priors was confirmed by a between model $\hat{R} \leq 1.05$ for Bull Trout and $\leq 1.12$ for Rainbow Trout.

### Recapture
The final CJS survival model estimated that the annual probability of being recaptured (and reported) by an angler ($\rho$) was 0.17 (95% CRI [0.11–0.23]) for Bull Trout (Table 6) and 0.14 (95% CRI [0.09–0.19]) for Rainbow Trout (Table 7). The indicator variable probabilities for $\gamma_{\rho Y}$ were 0.09 and 0.09 for each species, respectively, revealing little to no support for a change in the probability of recapture with year.

### Survival
The survival probability for Bull Trout was largely unaffected by spawning ($\gamma_{\phi\kappa} = 0.29$) but changed over the course of the study ($\gamma_{\phi Y} = 0.95$). More specifically, the annual Bull Trout survival probability was estimated to have declined from 0.91 (95% CRI [0.76–0.97]) in 2009 to just 0.46 (95% CRI [0.24–0.76]) in 2013 (Fig. 5).

In contrast, Rainbow trout survival was strongly affected by spawning ($\gamma_{\phi\kappa} = 1.00$) and may have changed over the course of the study ($\gamma_{\phi Y} = 0.76$). If present, the change in survival over time was a decline (Fig. 5). The CJS model estimated that spawning reduced the annual Rainbow Trout survival probability from 0.77 (95% CRI [0.68–0.85]) to 0.41 (95% CRI [0.30–0.53]) (Fig. 6).

### Spawning
For both species, the probability of spawning was clearly dependent on the fork length as $\gamma_{\kappa L} = 0.97$ for Bull Trout and 1.00 for Rainbow Trout. Figure 7 indicates that the probability of spawning increases from 0.40 (95% CRI [0.16– 0.75]) for a 500 mm Bull Trout to 0.94 (95% CRI [0.78–0.99]) for a 800 mm Bull Trout. The equivalent values for Rainbow Trout are 0.00 (95% CRI [0.00–0.00]) and 0.90 (95% CRI [0.76–0.98]), respectively.

There was little support ($\gamma_{\kappa Y} = 0.23$) for a change in the probability of spawning for Bull Trout from 2009 to 2013. However, in contrast, there was strong support ($\gamma_{\kappa Y} = 1.00$) for a decline in the probability of spawning for Rainbow Trout over the course of the study (Fig. 8).

**Table 6 Key fixed parameter estimates for the final Bull Trout CJS survival model.** The columns are the *parameter* name, the point *estimate*, the *lower* and *upper* 95% credible intervals, the standard deviation (*SD*), the percent relative *error*, and the statistical *significance*.

| Parameter | Estimate | Lower | Upper | SD | Error | Significance |
|---|---|---|---|---|---|---|
| $\beta_{\delta 0}$ | 2.358 | 1.987 | 2.772 | 0.200 | 17 | 0.0001 |
| $\beta_{\delta S}$ | −2.399 | −2.957 | −1.864 | 0.279 | 23 | 0.0001 |
| $\beta_{\delta Y}$ | −0.3732 | −0.9867 | 0.0342 | 0.3290 | 140 | 0.2642 |
| $\beta_{\kappa 0}$ | 2.429 | 1.192 | 3.589 | 0.589 | 49 | 0.0001 |
| $\beta_{\kappa L}$ | 1.147 | 0.027 | 1.970 | 0.433 | 85 | 0.0248 |
| $\beta_{\kappa Y}$ | −0.1275 | −1.1673 | 0.1006 | 0.3408 | 500 | 0.8014 |
| $\beta_{\phi \kappa}$ | 0.3302 | −0.3992 | 3.4160 | 0.9492 | 580 | 0.8190 |
| $\beta_{\phi 0}$ | 2.728 | 2.296 | 3.195 | 0.227 | 16 | 0.0001 |
| $\beta_{\phi Y}$ | −0.9816 | −1.6406 | −0.0110 | 0.3730 | 83 | 0.0368 |
| $\beta_{\rho 0}$ | −3.065 | −3.479 | −2.688 | 0.202 | 13 | 0.0001 |
| $\beta_{\rho Y}$ | 0.01149 | −0.08315 | 0.26485 | 0.09349 | 1500 | 0.9466 |
| $\gamma_{\delta S}$ | 1.000 | 1.000 | 1.000 | 0.000 | 0 | 0.0001 |
| $\gamma_{\delta Y}$ | 0.6408 | 0.0000 | 1.0000 | 0.4798 | 78 | 0.0001 |
| $\gamma_{\kappa L}$ | 0.9657 | 0.0000 | 1.0000 | 0.1820 | 52 | 0.0001 |
| $\gamma_{\kappa Y}$ | 0.2291 | 0.0000 | 1.0000 | 0.4203 | 220 | 0.0001 |
| $\gamma_{\phi \kappa}$ | 0.2902 | 0.0000 | 1.0000 | 0.4539 | 170 | 0.0001 |
| $\gamma_{\phi Y}$ | 0.9538 | 0.0000 | 1.0000 | 0.2099 | 52 | 0.0001 |
| $\gamma_{\rho Y}$ | 0.09250 | 0.00000 | 1.00000 | 0.28970 | 540 | 0.0001 |

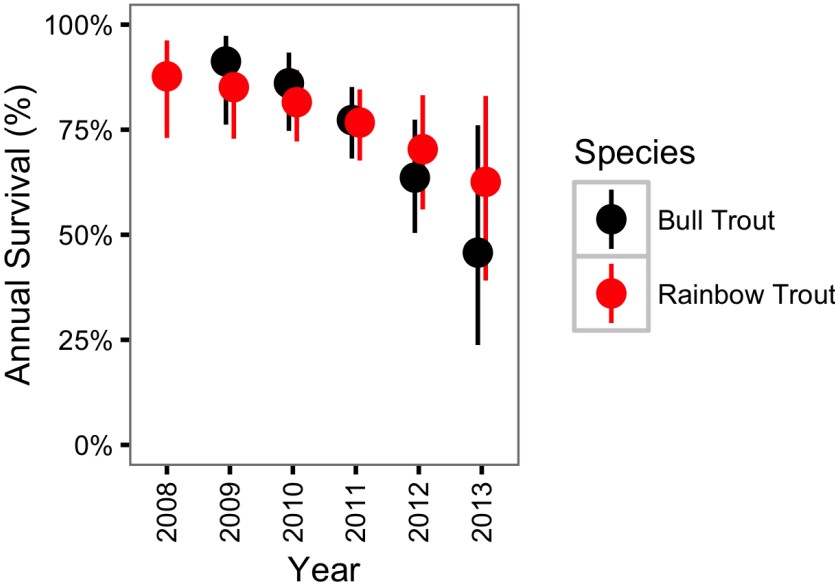

**Figure 5 Estimated annual survival probabilities for non-spawning Bull Trout and Rainbow Trout by year.** The bars represent 95% CRIs.

**Table 7** **Key fixed parameter estimates for the final Rainbow Trout CJS survival model.** The columns are the *parameter* name, the point *estimate*, the *lower* and *upper* 95% credible intervals, the standard deviation (*SD*), the percent relative *error*, and the statistical *significance*.

| Parameter | Estimate | Lower | Upper | SD | Error | Significance |
|---|---|---|---|---|---|---|
| $\beta_{\delta 0}$ | 2.975 | 2.543 | 3.502 | 0.242 | 16 | 0.0001 |
| $\beta_{\delta S}$ | −0.08340 | −0.91220 | 0.10070 | 0.25880 | 610 | 0.8392 |
| $\beta_{\delta Y}$ | −0.004190 | −0.234280 | 0.158230 | 0.099260 | 4,700 | 0.9886 |
| $\beta_{\kappa 0}$ | 0.1600 | −0.8004 | 1.1863 | 0.5032 | 620 | 0.7558 |
| $\beta_{\kappa L}$ | 4.457 | 2.876 | 6.296 | 0.874 | 38 | 0.0001 |
| $\beta_{\kappa Y}$ | −2.153 | −3.504 | −0.963 | 0.651 | 59 | 0.0010 |
| $\beta_{\phi \kappa}$ | −2.671 | −3.393 | −1.965 | 0.366 | 27 | 0.0001 |
| $\beta_{\phi 0}$ | 2.697 | 2.277 | 3.154 | 0.225 | 16 | 0.0001 |
| $\beta_{\phi Y}$ | −0.4681 | −1.0209 | 0.0283 | 0.3196 | 110 | 0.1818 |
| $\beta_{\rho 0}$ | −3.289 | −3.715 | −2.895 | 0.210 | 12 | 0.0001 |
| $\beta_{\rho Y}$ | 0.01512 | −0.06594 | 0.27639 | 0.08644 | 1,100 | 0.9008 |
| $\gamma_{\delta S}$ | 0.1963 | 0.0000 | 1.0000 | 0.3972 | 260 | 0.0001 |
| $\gamma_{\delta Y}$ | 0.09930 | 0.00000 | 1.00000 | 0.29910 | 500 | 0.0001 |
| $\gamma_{\kappa L}$ | 1.000 | 1.000 | 1.000 | 0.000 | 0 | 0.0001 |
| $\gamma_{\kappa Y}$ | 0.9983 | 1.0000 | 1.0000 | 0.0412 | 0 | 0.0001 |
| $\gamma_{\phi \kappa}$ | 1.000 | 1.000 | 1.000 | 0.000 | 0 | 0.0001 |
| $\gamma_{\phi Y}$ | 0.7645 | 0.0000 | 1.0000 | 0.4243 | 65 | 0.0001 |
| $\gamma_{\rho Y}$ | 0.08730 | 0.00000 | 1.00000 | 0.28230 | 570 | 0.0001 |

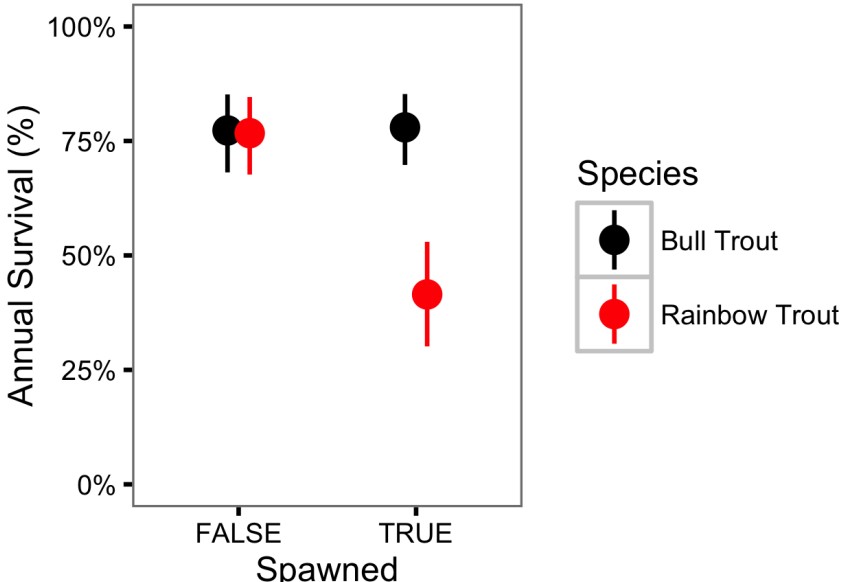

**Figure 6** **Estimated annual survival probabilities for Bull Trout and Rainbow Trout in 2011 by spawning status.** The bars represent 95% CRIs.

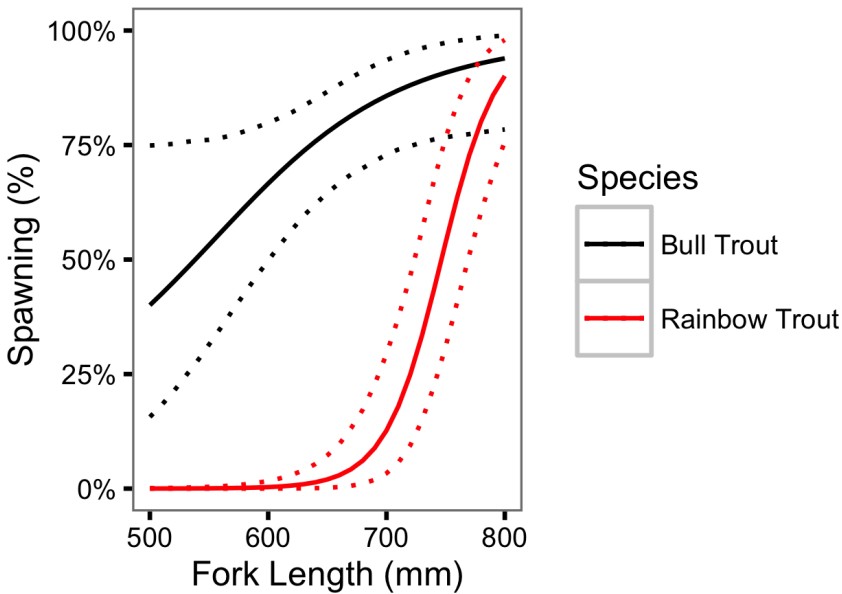

**Figure 7 Estimated probability of spawning for Bull Trout and Rainbow Trout by fork length.** The dotted lines represent 95% CRIs.

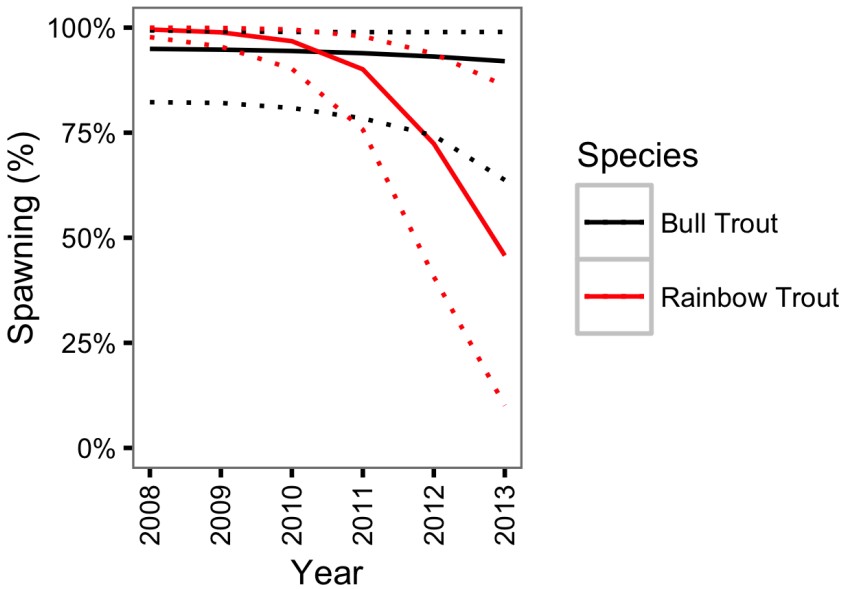

**Figure 8 Estimated probability of spawning for a 800 mm Bull Trout and Rainbow Trout by year.** The dotted lines represent 95% CRIs.

## Movement

The last key parameter in the CJS model—the probability of being detected moving among sections while alive with an active transmitter—varied by spawning season for Bull Trout ($\gamma_{\delta S} = 1.00$) but not Rainbow Trout ($\gamma_{\delta S} = 0.20$). In the case of Bull Trout, the probability was 0.49 (95% CRI [0.40–0.58]) for being detected moving during the summer spawning
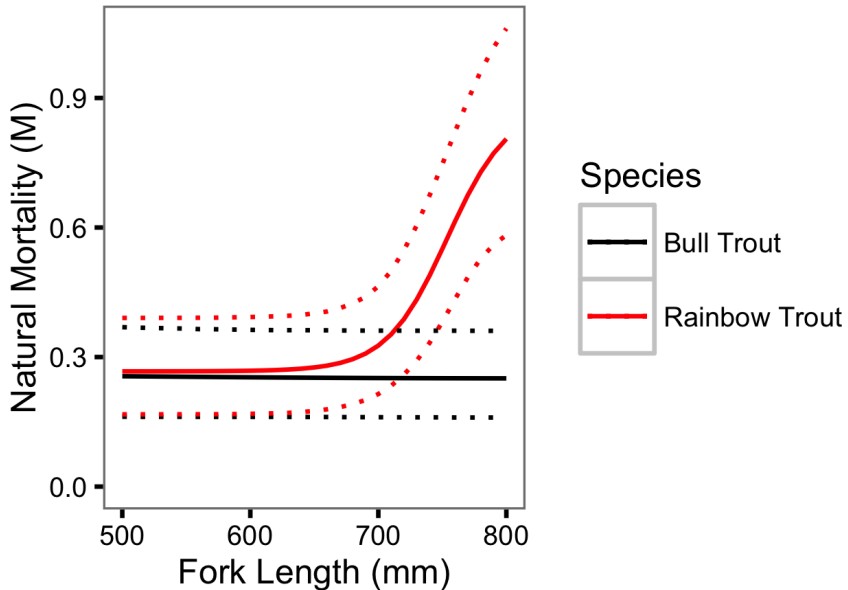

**Figure 9** **Estimated annual instantaneous natural mortality for Bull Trout and Rainbow Trout in 2011 by fork length.** The dotted lines represent 95% CRIs.

season compared to 0.91 (95% CRI [0.88–0.94]) during the other seasons. The probability of being detected moving was 0.95 (95% CRI [0.93–0.97]) for Rainbow Trout. Over the course of the study, the probability of being detected moving among sections may have $(\gamma_{\delta Y} = 0.64)$ declined $(\beta_{\delta Y} = -0.37)$ for Bull Trout but not Rainbow Trout $(\gamma_{\delta Y} = 0.10)$.

## Fishing and natural mortality

If all recaptures are harvested then $F = -\log(1 - \rho)$ and the estimated annual instantaneous fishing mortality $(F)$ is 0.18 (95% CRI [0.12–0.26]) for Bull Trout and 0.15 (95% CRI [0.10–0.22]) for Rainbow Trout. Similarly, as $M = -\log(\phi)$ the estimated annual instantaneous natural mortality $(M)$ for a Bull Trout was 0.09 (95% CRI [0.27–0.03]) in 2009 and 0.78 (95% CRI [1.44–0.27]) in 2013. For Rainbow Trout $M$ was 0.88 (95% CRI [1.20–0.64]) for a spawner and 0.27 (95% CRI [0.39–0.17]) for a non-spawner. The natural mortality is plotted by fork length in Fig. 9.

## Catchability

The creel survey estimated that in 2011, a grand total of 189,457 angler hours (and 201,434 rod hours) were expended fishing Kootenay Lake (*Andrusak & Andrusak, 2012*), which corresponds to an angler effort $(E)$ of 4.88 angler-hr $ha^{-1}$ $yr^{-1}$. If the catchability coefficient $(q)$ is defined to be the probability of (re)capture $(\rho)$ per unit of effort (*Ward et al., 2013*), i.e.,

$$q = \frac{\rho}{E} \tag{19}$$

then, in Kootenay Lake in 2011, $q = 0.034$ (95% CRI [0.023– 0.047]) angler-hr$^{-1}$ ha$^{-1}$ yr$^{-1}$ for Bull Trout and 0.028 (95% CRI [0.019–0.040]) angler-hr$^{-1}$ ha$^{-1}$ yr$^{-1}$ for Rainbow Trout.

## Density

The creel survey estimated that 4,845 large ($\geq$500 mm) Bull Trout and 3,320 large Rainbow Trout were caught by anglers (*Andrusak & Andrusak, 2012*). Given the estimated recapture rates, the catches suggest that the density of large Bull Trout in Kootenay Lake in 2011 was 0.75 fish ha$^{-1}$ and the density of large Rainbow Trout was 0.63 fish ha$^{-1}$.

# DISCUSSION

## Model reliablity

The reliability of any model's estimates depends on the extent to which its assumptions are met. If none of the CJS model's assumptions are violated then, the results indicate that the potential fishing mortality, if anglers harvest all fish, was $F = 0.18$ for Bull Trout and $F = 0.15$ for Rainbow Trout; that the natural mortality increased from $M = 0.09$ to $M = 0.78$ over the course of the study for Bull Trout; and that for Rainbow Trout, spawning increased $M$ from 0.27 to 0.88. The following paragraphs discuss each of the assumptions of the CJS model together with the implications of any violations for the mortality estimates.

*Sample.* The model assumes that the tagged individuals are a random sample from the population. The distribution of captures shows that fish were not sampled from all areas equally (Figs. S1–S2). However, the detections of acoustically tagged fish (Figs. S3–S4) indicated a high degree of mixing between areas. The possibility that fish may have differed in their susceptibility to (re)capture by angling is discussed below.

*Tagging.* The second assumption is that there are no effects of tagging. To reduce the chances of post-release mortality inflating $M$, fish that were no longer detected moving between sections 30 days after release were excluded from the analysis (*Hightower, Jackson & Pollock, 2001*). The fact that 7% of the Bull Trout versus 25% of Rainbow were judged to be post-release mortalities suggests that in the short-term at least Bull Trout are less affected by capture, handling and surgery. To ensure any longer-term effects of tagging were minimal, the transmitter to body weight ratio was $\leq 0.9\%$ (*Brown et al., 1999*). Nonetheless, it is possible that tagging could have increased natural or fishing mortality or reduced the probability of spawning due to infection or increased energetic expenditures.

*Anchor tag and transmitter loss.* The CJS model assumed that there was no anchor tag or transmitter loss (*Brenden, Jones & Ebener, 2010*). To minimize anchor tag loss, all fish were double-tagged. Analysis of double versus single-tagged recaptures indicates that the probability of losing both anchor tags was 4% for Bull Trout and 0.6% for Rainbow Trout. The higher loss rate for Bull Trout may be due to the spawning migration they undertake up steep turbulent tributaries (*Andrusak & Andrusak, 2014*). The exclusion of fish that had ceased moving within 30 days of initial capture reduced the chances that transmitter loss would affect the mortality estimates (*Hightower, Jackson & Pollock, 2001*).

*Identification.* Anglers were required to return all tags. Consequently it is unlikely that fish were misidentified.

*Emigration.* The model assumed that there was no movement outside of the main lake other than temporary seasonal spawning migrations by Bull Trout in the summer and Rainbow Trout in the spring. Bull Trout spawn in tributaries to Kootenay Lake (*Andrusak & Andrusak, 2014*) as well as tributaries to Trout Lake and Duncan Reservoir (*O'Brien, 2001*). The seven Bull Trout that were last detected at the top of the North Arm (sections S07 and S09) in the spring (Table 4) may have been early-leaving spawners headed to Trout Lake or Duncan Reservoir (*O'Brien, 2001*) that failed to return. The implications of the misidentification of spawning events is discussed below. The 18 Rainbow Trout that were last detected in the spring at the top of the North Arm or at section S02 at the outflow of Trout Lake (Table 5) are accounted for as spawning mortalities by the model. With the exception of the seven spring Bull Trout migrants at the top of the North Arm there is no indication of any significant movement outside the main lake other than temporary seasonal spawning migrations that are accounted for by the model.

*Recapture.* A key assumption of all mark-recapture studies is that there are no unmodelled individual differences in the probability of (re)capture in each time interval (*Biro, 2013*). As is often the case, the reliance on a single capture method means it is not possible to test this assumption (*Biro, 2013*). Depending on whether any individual differences were fixed or learned the estimated potential fishing mortality would be an over or under-estimate, respectively (*Askey et al., 2006*; *Biro, 2013*).

*Survival.* The full model allowed survival to vary with spawning status and year. Simulations by *Matechou et al. (2013)* indicate that the survival estimates are likely to be robust to any remaining time-dependent variation.

*Independence.* Individuals exhibited no obvious shoaling or coordination of movements other than spawning migrations. Consequently the fate of each individual was likely independent of the fate of any other individual.

*Instantaneous.* While individuals were almost immediately released, the sampling periods for captures, recaptures and detections were effectively continuous. However, because the seasonal survival was high ($\geq$82% for Bull Trout and $\geq$80% for Rainbow Trout) and because the purpose was to estimate overall annual survival across four seasonal time periods, substantial biases are not expected (*Barbour, Ponciano & Lorenzen, 2013*).

*Movement.* The CJS model assumes that mortalities exhibit no movement. Consequently if the model was to overestimate the probability of a living individual with an active transmitter being detected moving between sections then $M$ would also be overestimated. However, this is unlikely for three reasons. Firstly, the model allowed the probability of movement to vary with spawning season and year. Secondly, the seasonal probability of detecting an individual moving between sections was relatively high (91% for Bull Trout and 95% for Rainbow Trout). Finally, the state-space component of the model means that a temporarily inactive individual or one that is subsequently recaptured will not be misidentified as a natural mortality. Consequently any bias in $M$ is expected to be small.

*Reporting.* To maximize the tag reporting rate, which was assumed to be 100%, at least one of the tags was worth $100 tag (*Bacheler et al., 2009*). Previous studies on Common Snook (*Centropomus undecimalis*) in Florida in the late 1990s (*Taylor et al., 2006*) and multiple freshwater fish species in Idaho from 2006 to 2009 (*Meyer et al., 2012*) suggest that $100 is sufficient to achieve reporting rates of approximately 99%.

*Spawning.* The model assumed that spawning events are correctly identified. Large Rainbow Trout from Kootenay Lake almost exclusively spawn at the head of the Lardeau River (*Irvine, 1978*; *Irvine, Baxter & Thorley, 2013*). As this spawning area was acoustically monitored duing the spring it is likely that most of the Rainbow Trout spawning events were correctly identified.

For Bull Trout the situation is more complicated. Large Bull Trout from Kootenay Lake spawn in multiple tributaries of Kootenay Lake, Trout Lake and Duncan Reservoir (*O'Brien, 2001*; *Andrusak & Andrusak, 2014*). Consequently it was not possible to monitor spawning locations and spawning was inferred by a hiatus in detections for four or more weeks during August and September. As a result only fish which successfully returned to Kootenay Lake in the fall post-spawning were identified as spawners. Bull Trout that succumbed to spawning-related injuries, stress or predation prior to returning to Kootenay Lake were identified as inlake natural mortalities As a result $M$ is likely biased high for non-spawning Bull Trout and biased low for spawners. This bias may explain why an effect of spawning on mortality was not detected for Bull Trout.

*Growth.* The model assumed that growth was described by a seasonal Von Bertalanffy Growth Curve with a $L_\infty$ of 1,000 mm and a $k$ of 0.13 for Bull Trout and a $k$ of 0.19 for Rainbow Trout. The curve for Bull Trout was partly based on large Bull Trout in Adams Lake, British Columbia (*Bison, O'Brien & Martell, 2003*). The growth curve for Rainbow Trout was based on lake specific data (*Andrusak & Thorley, 2014*) and is likely a good approximation.

### Mortality estimates

The implications of any violations of the assumptions are negligible for the mortality estimates with two exceptions. The reliance on a single capture method means that $F$ could be an over or under-estimate for either or both species. Similarly, the difficulty identifying Bull Trout spawners means that spawning Bull Trout may experience a higher $M$ than non-spawners.

### Fishing mortality

The actual fishing mortalities are dependent on the release and handling mortality rates. Taken at face value the estimates from the CJS model indicate that if anglers were to harvest all fish, then the annual instantaneous fishing mortality would be approximately $F = 0.18$ for Bull Trout and $F = 0.15$ for Rainbow Trout. The corresponding annual interval fishing mortalities are 17% and 14%, respectively. The creel survey calculated that in 2011 anglers released 40% of the Bull Trout and 62% of the Rainbow Trout they captured (*Andrusak & Andrusak, 2012*). Consequently, if large fish are released at the same rate as small fish and

there are no post-release mortalities, then the effective annual interval fishing mortalities would be approximately 10% for Bull Trout and 5% for Rainbow Trout.

### Post-release mortality

The current study indicates that the large Bull Trout in Kootenay Lake are relatively robust to angling and surgical implantation of an acoustic transmitter with a 30 day post-release mortality of 7%. For comparison, *Post et al. (2003)* in their age-structured model of an adfluvial bull trout population presumed post-release mortality was 10% but considered values of 2% and 25%. The current study estimated that large Rainbow Trout experience a post-release mortality of 25%. For comparison, meta-analyses indicate that the mean post-release mortality in salmonid recreational fisheries is 16–18% although the distribution is positively skewed with most estimates $\leq$ 11% (*Bartholomew & Bohnsack, 2005*; *Huhn & Arlinghaus, 2011*). It is worth noting that the current estimates include surgical implantation of a acoustic transmitter and may therefore be inflated relative to the post-release mortality of fish caught-and-released by recreational anglers. It is however also possible that the care taken playing, landing and recovering the fish resulted in a comparable or even lower post-release mortality rate than in the recreational fishery.

### Catchability

Remarkably few estimates of catchability have been published for large piscivorous trout. A notable exception is *Post et al. (2003)* who based on creel data from Lower Kananaskis Lake and experimental fishery data from Marie Lake and Quirk Creek considered a $q$ of 0.07 fish caught.vulnerable fish$^{-1}$ angler-hr$^{-1}$ ha$^{-1}$ yr$^{-1}$ to be representative for Bull Trout. They assumed fish were invulnerable to capture at $\leq$200 mm and completely vulnerable by 400 mm. Their nominal value of 0.07 is more than double the current Bull Trout estimate of 0.034 angler-hr$^{-1}$ ha$^{-1}$ yr$^{-1}$. However, *Post et al. (2003)* did consider the possibility that $q$ ranged between 0.035 and 0.14. The reasons for the relatively low catchability of Bull Trout in Kootenay Lake are unknown but could include less efficient angling methods (anglers primarily target Rainbow Trout) or more uniform fish distributions.

Catchability is typically strongly negatively related to the population density—a phenomenon known as hyperstability (*Ward et al., 2013*). Based on a meta-analysis of Lake Trout (*Salvelinus namaycush*) in 12 lakes in Ontario, Canada, *Shuter et al. (1998)* modelled the relationship between catchability and density ($D$) using a modified form of the equation

$$q = \exp\left(-\frac{0.14 \cdot E}{1 + 0.35 \cdot D}\right) \cdot E^{-1}. \tag{20}$$

In Kootenay Lake in 2011, Bull Trout occured at a density of 0.75 fish.ha$^{-1}$ and experienced an effort of 4.88 angler-hr ha$^{-1}$ yr$^{-1}$. If large Bull Trout have the same catchability as Lake Trout, Eq. (20) predicts a catchability of 0.12 angler-hr$^{-1}$ ha$^{-1}$ yr$^{-1}$. Although less then the upper range of 0.14 considered by *Post et al. (2003)*, 0.12 is still much higher than the estimated value of 0.034 angler-hr$^{-1}$ ha$^{-1}$ yr$^{-1}$.

*Ward et al. (2013)* concluded that hyperstability for stocked Rainbow Trout in 18 British Columbian lake was caused by differences in angler skill with less experienced anglers

tending to target lakes with higher fish densities. The lakes were all small waterbodies (<45 ha) with high fish densities (>15 fish ha$^{-1}$) and no natural recruitment. As the predictions were for small insectivorous trout at high densities they could not be reliably extrapolated to large piscivorous Rainbow Trout. Nonetheless, *Ward et al.*'s *(2013)* findings suggest that Catch-Per-Unit-Effort may need to be scaled by angler effort when attempting to model Bull Trout and Rainbow Trout abundance in Kootenay Lake.

### Inter-annual variability

There was little to no support for a change in the fishing mortality for either Bull Trout or Rainbow Trout over the course of the study. This finding is important because, as discussed below, it suggests that angler effort is not the primary driver of any short-term fluctuations.

## Natural mortality

*Post et al. (2003)* assumed a constant annual instantaneous natural mortality of 0.20 for their Bull Trout population model. The estimate was based on the observed mortality of mature Bull Trout in Lower Kananaskis Lake and *Shuter et al.*'s *(1998)* estimates for Lake Trout. *Shuter et al. (1998)* tabulated estimates of $M$ from 12 lakes. The largest value was 0.4 while the remaining 11 were between 0.12 and 0.25. For comparison, the CJS model estimated that the annual instantaneous natural mortality of Bull Trout increased from 0.09 in 2009 to 0.78 in 2013; although it is worth noting that the upper estimate in 2009 was 0.27 and the lower estimate in 2013 was 0.27.

The current study estimated that the annual instantaneous natural mortality of a non-spawning large Rainbow Trout was 0.27 compared to 0.88 for a spawner. The ecological, evolutionary and management implications of the mortality cost of spawning are discussed below. As the probability of spawning increased with body length the annual instantaneous natural mortality rate was also size-dependent.

To the best of our knowledge, a direct estimate (*Hoenig et al., 2016*) of the natural mortality of large Rainbow Trout has not been published in the peer-reviewed literature. There are however numerous empirical (indirect) equations for calculating $M$ based on other life-history parameters (*Then et al., 2015*; *Hamel, 2015*). Based on over 200 fish species, *Then et al. (2015)* recommended the maximum age ($t_{max}$) based estimator

$$M = 4.899 \cdot t_{max}^{-0.916} \tag{21}$$

when possible and the growth-based method

$$M = 4.188 \cdot k^{0.73} \cdot L_\infty^{-0.33} \tag{22}$$

otherwise. When applied to large Rainbow Trout in Kootenay Lake with $t_{max} = 8$ (*Andrusak & Andrusak, 2015*), $k = 0.19$ (*Andrusak & Thorley, 2014*) and $L_\infty = 1000$, the estimated values of $M$ are 0.73 using Eq. (21) and 0.13 using Eq. (22). The discrepancy in the empirical estimates as well as their inability to account for annual and length-based variation in $M$ indicates the limitations of an indirect approach to estimating the natural mortality of large piscivorous trout.
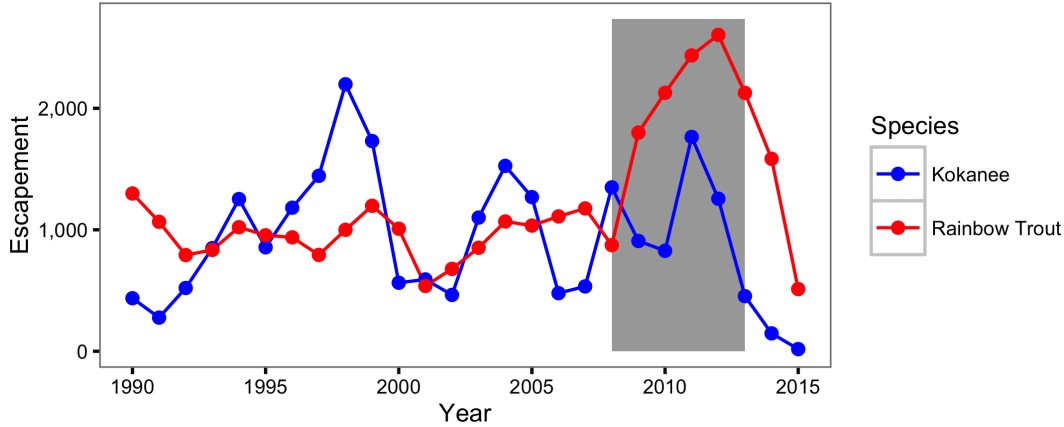

**Figure 10  Rainbow Trout and Kokanee escapement by year.** The Rainbow trout escapement is the spawner abundance at the outflow of Trout Lake. The Kokanee escapement is the spawner abundance at Meadow Creek and the Lardeau River. The grey rectangle indicates the period of the study from 2008 to 2013.

### Inter-annual variability

The CJS model strongly supported an increase in the natural mortality of Bull Trout over the course of the study. Further support for an increase in Bull Trout natural mortality is also provided by a preliminary redd count analysis for Kootenay Lake (*Andrusak & Andrusak, 2014*) that suggests a lake-wide decrease in the number of spawners from a peak in 2009 to a low in 2013 (the last year of the redd count study). As the CJS model provided little support for a change in the probability of Bull Trout spawning the decline in the number of Bull Trout spawners likely reflects a decline in Bull Trout abundance. The finding that $M$ has changed over the course of the study is important because it is consistent with Kokanee abundance or another biological factor as a driver of short-term Bull Trout population dynamics.

The CJS model only provided moderate support for an increase in the natural mortality of Rainbow Trout. More specifically the CJS model estimated that the annual instantaneous natural mortality of a non-spawning Rainbow Trout increased from 0.13 in 2008 to 0.47 in 2013. In contrast, Rainbow Trout escapement, as estimated by the peak count with a multiplier of 3.08 (*Hagen et al., 2010*), increased from 1,583 in 2008 to a high of 3,289 in 2012 before falling slightly in the spring of the final year of the current study (Fig. 10). While the escapement was increasing to record levels the probability of spawning was declining dramatically. This finding is important because it suggests that the actual increase in the number of large Rainbow Trout was even more dramatic than indicated by the escapement.

### Spawning

For Bull Trout the CJS model did not identify an increase in the mortality rate associated with spawning. However, as discussed above spawners which died prior to returning to Kootenay Lake would have been misidentified as inlake mortalities. It is therefore likely

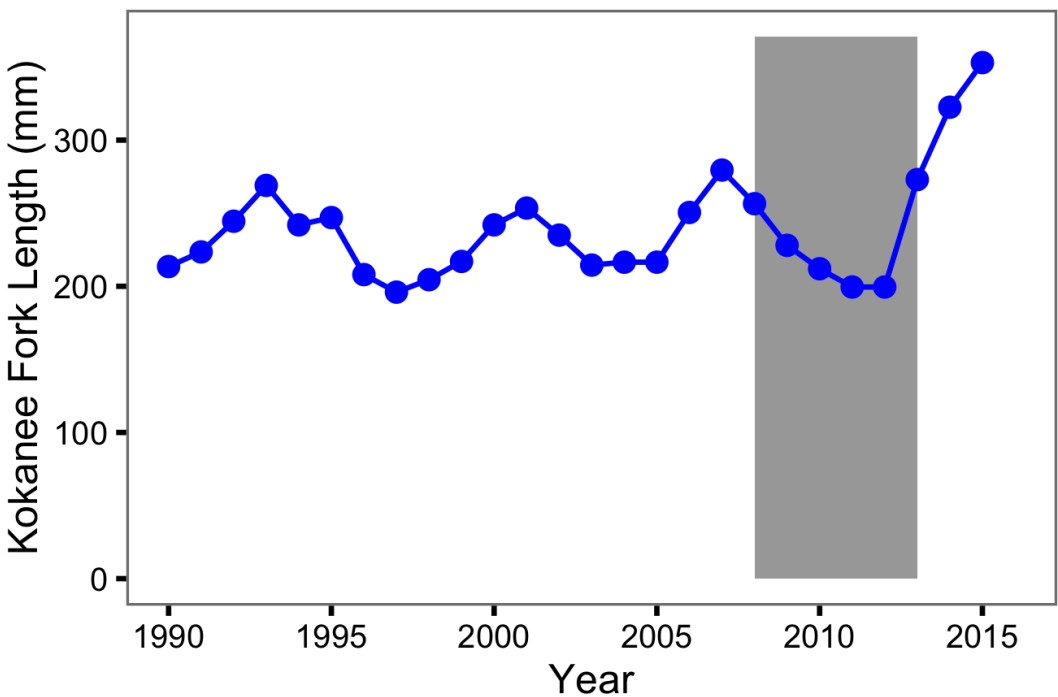

**Figure 11  Kokanee spawner length by year.** The lengths are the mean fork length at Meadow Creek Spawning Channel. The grey rectangle indicates the period of the study from 2008 to 2013.

that the natural mortality of spawners and non-spawners are under and over-estimated, respectively.

For Rainbow Trout, spawning increased the annual interval natural mortality rate from 23% to 59%. The high spawning mortality is consistent with the limited spawning area and high levels of antagonistic interactions at the outflow of Trout Lake (*Hartman, 1969*; *Hartman & Galbraith, 1970*). The high spawning mortality may also explain why the probability of spawning is low for Kootenay Lake Rainbow Trout smaller than approximately 650 mm in length, i.e., the cost of spawning selects for delayed maturation.

## Population fluctuations

There was no evidence for a change in the fishing mortality over the course of the study but strong and moderate evidence for an increase in the natural mortality of Bull Trout and Rainbow Trout, respectively. These results are consistent with Kokanee or another biological factor, as opposed to anglers, as the driver of the short-term fluctuations in the Rainbow Trout population. Evidence for a role of Kokanee is provided by the numbers (Fig. 10) and sizes (Fig. 11) of Kokanee spawners at the Meadow Creek spawning channel and the Lardeau River (*MFLNRO, 2016*).

An additional key finding is that the probability of spawning declined dramatically for Rainbow Trout over the course of the study. Given the high mortality cost of spawning, any reduction in the energy available for gamete production associated with the decline in Kokanee abundance might cause fish to delay spawning. An important implication of this potential relationship is that Rainbow Trout escapement at the outflow to Trout Lake

may be a less reliable index of population abundance than previously assumed. Perhaps due to their density-dependent phenotypic plasticity the probability of spawning remained constant for Bull Trout (*Johnston & Post, 2009*).

It is important to note that the results of the current study do not preclude changes in angler effort or lake productivity as drivers of longer-term trends in abundance. Indeed in a recent modelling study *Kurota et al. (2016)* concluded that the peak in Rainbow Trout in 2012 was caused by a longer-term increase in recruitment combined with a longer-term reduction in $F$.

Based on the recapture rates and creel survey catch estimates, the current study found that in 2011 large Bull Trout were at least as abundant as large Rainbow Trout in Kootenay Lake. This finding is important because it means that multi-species population models such as that developed by *Kurota et al. (2016)* need to be expanded to take Bull Trout into account.

## ACKNOWLEDGEMENTS

We thank Robyn Irvine (Poisson Consulting), Harvey Andrusak (Redfish Consulting), Matt Neufeld (MFLNRO), Jeff Burrows (MFLNRO), James Baxter (FWCP) and others for assisting with fish capture and tagging; Kerry Reed (Reel Adventures) and Alan Richardson (Magic Charters) for providing guide services; Kerry Reed for tagging the fish in 2012 and 2013; Sarah Stephenson (MFLNRO) and Matt Neufeld for maintaining and downloading the acoustic receiver array; all the anglers who reported recaptures; MFLNRO staff for recording the reported recaptures; Gary Pavan for processing the detection and capture data and creating the lake sections; Bernhard Konrad for providing code to check the consistency of detections; and Harvey Andrusak, Steve Arndt (MFLNRO), Crystal Klym (FWCP) and Robyn Irvine for providing comments on earlier drafts. We particularly thank Jeff Burrows and Matt Neufeld for supporting the project.

The Habitat Conservation Trust Foundation was created by an act of the legislature to preserve, restore and enhance key areas of habitat for fish and wildlife throughout British Columbia. Anglers, hunters, trappers and guides contribute to the projects of the Foundation through licence surcharges. Tax deductible donations to assist in the work of the Foundation are also welcomed. The acoustic receiver array is a trans-boundary partnership between the Ministry of Forests, Lands and Natural Resource Operations (MFLNRO), with capital funding provided by the US Fish and Wildlife Service (USFWS) and the Bonneville Power Administration (BPA) through the Northwest Power and Conservation Council's Fish and Wildlife Program. Annual operation and maintenance of the array was completed by MFLNRO and funded by FWCP in conjunction with BPA through the Northwest Power and Conservation Council's Fish and Wildlife Program, in cooperation with the Idaho Department of Fish and Game (IDFG), and the Kootenai Tribe of Idaho (KTOI).

### Funding

The project was primarily funded by the Habitat Conservation Trust Foundation (HCTF). The project was also partially funded by the Fish and Wildlife Compensation Program

(FWCP) on behalf of its program partners BC Hydro, the Province of B.C., Fisheries and Oceans Canada (DFO), First Nations and the public who work together to conserve and enhance fish and wildlife impacted by the construction of BC Hydro dams. The Freshwater Fish Society of British Columbia (FFSBC) provided the tag rewards. The funders had no role in study design, data collection and analysis, decision to publish, or preparation of the manuscript.

### Grant Disclosures

The following grant information was disclosed by the authors:
Habitat Conservation Trust Foundation (HCTF).
Fish and Wildlife Compensation Program (FWCP).

### Competing Interests

Joseph Thorley is employed by Poisson Consulting Ltd. to provide independent analytic services for a wide variety of government agencies, corporations and conservation organizations on a range of different species and issues.

### Author Contributions

- Joseph L. Thorley conceived and designed the experiments, performed the experiments, analyzed the data, wrote the paper, prepared figures and/or tables.
- Greg F. Andrusak conceived and designed the experiments, performed the experiments, wrote the paper.

### Animal Ethics

The following information was supplied relating to ethical approvals (i.e., approving body and any reference numbers):

Fish were obtained under scientific collection permits (CB08-43988, CB09-53420, CB10-61021, CB11-69505, CB12-76723) issued by the British Columbia Ministry of Forests Lands and Natural Resource Operations (MFLNRO).

### Data Availability

klexdatr R data package
https://github.com/Poissonconsulting/klexdatr
https://doi.org/10.5281/zenodo.162982
lexr R package
https://github.com/Poissonconsulting/lexr
https://doi.org/10.5281/zenodo.163003
klexr R package
https://github.com/Poissonconsulting/klexr
https://doi.org/10.5281/zenodo.163648.

### Supplemental Information

Supplemental information for this article can be found online at http://dx.doi.org/10.7717/peerj.2874#supplemental-information.

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
