# Peer review of "The fishing and natural mortality of large, piscivorous Bull Trout and Rainbow Trout in Kootenay Lake, British Columbia (2008–2013)"

_PeerJ, doi:10.7717/peerj.2874_

## Round 0.1 · original submission · Minor Revisions

Please respond to the comments of the two reviewers.

·

Basic reporting

no comments

Experimental design

no comments

Validity of the findings

I feel that a significant limitation of the study is the lack of comparable data for resident Kokanee populations. In several places, it is suggested that changes in Kokanee numbers have resulted in declines of trout, yet the authors provide no concrete data to support this. In fact, the data provided for Kokanee spawners does not show any relationship with Rainbow Trout for the years 1990-2010, yet this is entirely ignored. This should be discussed. It may be that Kokanee are a driver for Trout populations, but I do not believe that the authors have provided any strong data to support this. At the very least, I believe that this omission (limitation) should be stated up-front in the MS, and reiterated near the end of the Discussion section (i.e., Management Implications).

Additional comments

Well written and described paper, with substantial data on Rainbow Trout and Bull Trout populations in Kootenay Lake. I was surprised not to see any mention of the natural history of Kootenay Lake (i.e., it is a human engineered lake system, with consequences for aquatic productivity). I’d like to see a few sentences included in key sections of the MS. When was the dam (dams?) built, and how might that have altered fish populations and aquatic productivity.

Also (unless I missed it), the authors fail to suggest why it is that Rainbow Trout survival is associated with spawning, but not Bull Trout. It is a key finding of their work, and I for one am interested to know why there is this apparent difference. This should be included.

Reviewer 2 ·

Basic reporting

The manuscript is well-written and the results are presented in a very clear fashion. There is sufficient referencing of the literature that this study can be placed in context of similar studies elsewhere. My only suggestion for improvement of the presentation is to combine figures 10 and 11 by overlaying the changes in kokanee abundance on the trout abundance.

Experimental design

This is a very well executed study, done on an interesting and socially important lake. The overall question is fairly mundane in the grand scheme of things, but the results that are specific to this ecosystem are clear and will be important for management of these resources.

While the effects of size on survival rates, probability of spawning, etc. are given relative to the length of the fish, it would be very useful to provide a size-at-age function in this manuscript in case other scientists would like to use these results as a function of age.

I do not understand why catchability is expressed on a per ha basis. I would simply express these results on a lake basis (ie probability of capture per rod*day). While mathematically correct to express these per ha, it makes little practical sense to do so.

Validity of the findings

The development and application of the model seem valid and the discussion of assumptions is thorough (though maybe a bit dull). The results are clearly valid given the experimental design and the treatment of the data. I think it is worth a brief discussion of how the sizes of kokanee have changed during their recent crash. Is there any evidence of density-dependence in growth of kokanee? There should be and this response should be noted because both size of kokanee and their density will affect the potential feeding rate of the trout. It is certainly possible that kokanee can grow out of the window of vulnerability to predation by trout when they are found at low densities in the lake. I encourage some speculation here.

On line 514 it is stated that there was a 'decline in natural mortality'. Is this correct. I think there was a decline in survival (ie an increase in natural mortality).

Additional comments

Nice paper overall.

---

## Round 0.2 · accepted · Accept

The revision was credibly responsive to the reviews which smoothed out the paper quite nicely. The paper is now complete but an interesting point is raised by reviewer 1 (Price); that is, is the fertilization program on Kootenay Lake actually influential on the population cycles given in this paper. I'd be interested in a direct communication from you about this issue including any papers on the subject that you may know about: [email protected].